# SpecVLM: Fast Speculative Decoding in Vision-Language Models

## Abstract

Speculative decoding is a powerful way to accelerate autoregressive large language models (LLMs), but directly porting it to vision-language models (VLMs) faces unique systems constraints: the prefill stage is dominated by visual tokens whose count scales with image resolution and video length, inflating both compute and memory—especially the key-value (KV) cache. We study speculative decoding for VLMs and introduce **SpecVLM**, a practical system that (1) establishes a strong EAGLE-2-style baseline, *EagleVLM*, delivering 1.5–2.3× end-to-end speedups over full autoregressive inference, and (2) further accelerates VLM inference with an **elastic visual compressor** that adaptively selects among pruning, pooling, convolution, and resampler primitives to balance FLOPs/parameters and accuracy per input. To avoid costly offline distillation corpora, we propose an **online-logit distillation** protocol that trains the draft model with on-the-fly teacher logits and penultimate features using a combined cross-entropy and Smooth L1 objective, eliminating storage and preprocessing while remaining compute-efficient. This protocol reveals a **training-time scaling** effect: longer online training monotonically increases the draft model's average accepted length, improving speculative efficiency. Empirically, SpecVLM achieves additional acceleration, culminating in **2.5–2.9× end-to-end speedups within 5 epochs** across LLaVA and MMMU, consistently over resolutions and task difficulties, while preserving the target model's output distribution (lossless decoding).

## 1 Introduction

Autoregressive decoding underpins many high-quality vision-language models (VLMs) such as LLaVA (Liu et al., 2023), GPT-4 (Achiam et al., 2023), and Gemini (Team et al., 2023), which are widely used for image captioning, visual question answering, and multimodal dialogue. While these teacher models produce high-fidelity outputs, their token-by-token decoding is computationally expensive—an issue that is amplified in multimodal settings because the prefill stage (visual encoding → projection → token injection) can dominate wall-clock time and memory usage (Li et al., 2025b). Higher image resolutions, denser visual tokenizations, and video inputs dramatically increase the number of visual tokens, which in turn inflates both the **prefill** cost and the **KV cache traffic** during decoding. This growth undermines throughput and latency, complicating real-time and large-scale deployment.

We first conduct a comprehensive latency breakdown of the LLaVA-1.6-7B (Liu et al., 2024) model (batch size=1, temperature=0), as illustrated in Figure 1(a). The analysis shows that, beyond the inevitable latency from the vision encoder and projector, the LLM prefill stage is the dominant bottleneck. The remaining latency stems from next-token prediction during decoding. Although each decoding step is relatively lightweight compared to prefill, the autoregressive process accumulates, forming a second critical bottleneck.

To tackle these dual bottlenecks, we combine two complementary accelerations. First, we apply speculative decoding (Leviathan et al., 2023) to reduce the number of autoregressive steps, improving latency while preserving the target model's output distribution (lossless decoding). Second, we integrate visual token compression (Li et al., 2024a) and propose an elastic, question-adaptive compressor to reduce the number of visual tokens. This alleviates both compute and memory usage in the draft model prefill and reduces KV-cache traffic during decoding, as shown in Figure 1(b).

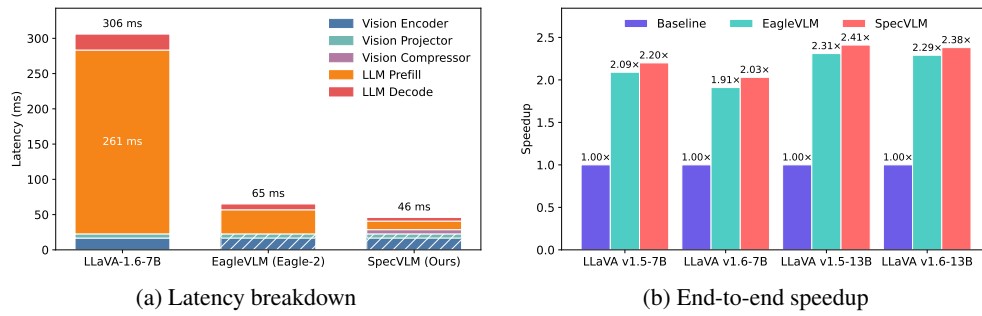

(a) Latency breakdown

(b) End-to-end speedup

Figure 1: Accelerating LLaVA models with speculative decoding and adaptive visual token compression (batch size=1, temperature=0). (a) Latency breakdown on NVIDIA RTX 4090. (b) End-to-end speedup for LLaVA-v1.5 and LLaVA-v1.6 on the LLaVA-Bench-in-the-Wild benchmark.

Speculative decoding—running a smaller, faster *draft* model to propose multiple tokens and verifying them with a heavier *target* model—has proven effective for text-only LLMs. However, porting this to VLMs introduces unique challenges (see Figure 2). **First**, the draft must process visual inputs efficiently during prefill; naively reusing LLM drafts leaves the draft burdened by large visual token sets. **Second**, our survey shows four major classes of visual token compressors—pruning (Shang et al., 2024; Chen et al., 2024; Huang et al., 2024; Zhang et al., 2024; Cao et al., 2024; Alvar et al., 2025; Chen et al., 2025; Tao et al., 2025; Ye et al., 2025a; Lin et al., 2025; Ye et al., 2025b), pooling (Cha et al., 2024; Li et al., 2024b; Cai et al., 2024a), convolutional (Chu et al., 2023; Dong et al., 2024), and resampler-based (Li et al., 2023; Fan et al., 2024; Hu et al., 2024; Li et al., 2024a; 2025a; Zhang et al., 2025)—each with complementary strengths and weaknesses. Parameter-free methods (pruning, pooling) are cheap but less expressive; learned resamplers and convolutional compressors are more expressive but add parameters and compute. There has been little exploration of **how to select or combine these compressors inside a speculative decoding pipeline to match task difficulty, model capacity, and compute budgets**.

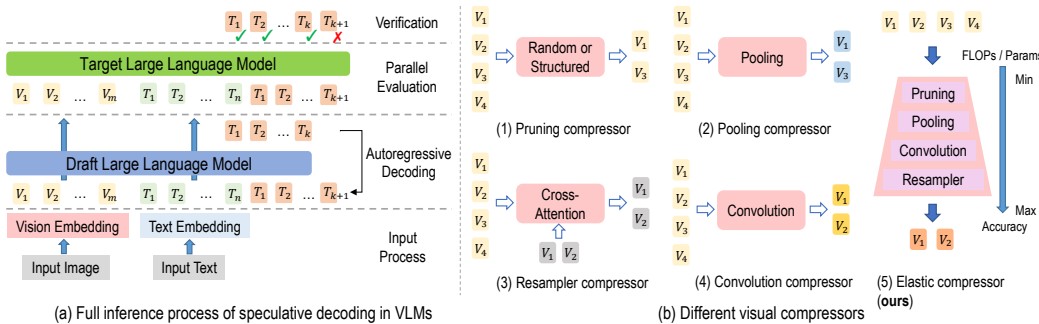

(a) Full inference process of speculative decoding in VLMs

(b) Different visual compressors

Figure 2: Speculative decoding for VLMs (Gagrani et al., 2024) and visual compression strategies. Our elastic approach yields favorable speed-accuracy trade-offs.

We take a comprehensive approach. We first construct EagleVLM, a strong EAGLE-2-inspired speculative baseline for VLMs that integrates a faithful draft with the full target model, demonstrating 1.5–2× speedups over end-to-end autoregressive inference. Building on this baseline, we introduce two key innovations to further reduce VLM inference cost: (1) an **elastic visual compressor** that dynamically selects pruning, pooling, resampler, and convolution operators to meet accuracy-compute trade-offs per input; and (2) an **online-logit distillation** protocol that trains the draft with on-the-fly teacher supervision, avoiding large offline distillation corpora while improving alignment.

An empirical finding is particularly noteworthy: with fixed training data and draft architecture, increasing online training time monotonically increases the draft model's *average accepted length*, thereby improving speculative efficiency. This *training-time scaling* effect suggests multimodal speculative decoding benefits from longer, targeted training regimes beyond what LLM-only studies have reported. Finally, our SpecVLM—combining the elastic compressor with online-logit train-

ing—delivers overall end-to-end acceleration up to 2.5–2.9× within 5 epochs, without sacrificing output fidelity. Our contributions are summarized as follows:

- We conduct a systematic study of speculative decoding applied to VLMs and a strong EAGLE-2-style baseline (EagleVLM) for that setting.

- We further propose SpecVLM, an elastic visual compressor that adaptively trades off compute and accuracy by selecting appropriate compression primitives.

- An online-logit distillation protocol for efficient draft training that eliminates the need for large offline distillation datasets and reveals a training-time scaling-up effect.

- We validate SpecVLM on the LLaVA benchmark suite and the MMMU multimodal evaluation sets, demonstrating consistent and robust performance across different model sizes and task difficulties.

## 2    RELATED WORK

### 2.1    SPECULATIVE DECODING

Speculative decoding accelerates autoregressive models by letting a smaller, faster draft propose multiple tokens that a larger target verifies in parallel (Leviathan et al., 2023; Chen et al., 2023). Early work explored greedy blockwise acceptance (Stern et al., 2018). Systems such as SpecInfer (Miao et al., 2023), Medusa (Cai et al., 2024b), and Hydra (Ankner et al., 2024) improve efficiency via parallel verification and auxiliary heads, but can suffer from limited token diversity and coupling across steps. EAGLE (Li et al., 2024c;d) addresses this by autoregressing at the feature level and employing static/dynamic tree-structured drafts that decouple time steps. Jakiro (Huang et al., 2025a) further enhances diversity with Mixture-of-Experts (MoE) heads at the cost of added latency. However, these approaches largely assume text-only models and overlook VLM-specific constraints: heavy visual prefill, large KV caches, and resolution-/video-driven sequence growth. They also typically depend on offline teacher-logit corpora, which are cumbersome at multimodal scales. Recent works have explored speculative decoding in VLMs through SPD-MLLM (Gagrani et al., 2024) and Spec-LLaVA (Huo et al.), but these rely on carefully designed standalone LLM drafts requiring significant architectural modifications. TwigVLM (Shao et al., 2025) combines speculative decoding with pruning techniques but produces lossy results. Additionally, mainstream speculative decoding approaches rely on offline dataset generation (Li et al., 2024c;d; Cai et al., 2024b; Ankner et al., 2024; Huang et al., 2025a), which can be storage-intensive and computationally expensive for large VLMs. In contrast, our SpecVLM uses only a single transformer decoder layer combined with online logits distillation, eliminating the cumbersome data preprocessing.

### 2.2    VISUAL TOKEN COMPRESSION FOR VLMS

Visual token compression is central to efficient VLM inference with high-resolution images and long visual sequences. Methods fall into four classes: (1) **Pruning** removes redundant tokens (e.g., LLaVA-PruMerge (Shang et al., 2024), SparseVLM (Zhang et al., 2024), DivPrune (Alvar et al., 2025), FitPrune (Ye et al., 2025a), ATP-LLaVA (Ye et al., 2025b), DyCoke (Tao et al., 2025), Recoverable Compression (Chen et al., 2025)); (2) **Pooling** aggregates local features (Honeybee (Cha et al., 2024), LLaMA-VID (Li et al., 2024b), Matryoshka (Cai et al., 2024a)); (3) **Convolution** applies learned downsampling (MobileVLM (Chu et al., 2023), InternLM-XComposer2-4KHD (Dong et al., 2024)); and (4) **Resampler** reallocates tokens via cross-attention (BLIP-2 (Li et al., 2023), MoUSI (Fan et al., 2024), Matryoshka (Hu et al., 2024), TokenPacker (Li et al., 2025a), LLaVA-Mini (Zhang et al., 2025)). Parameter-free approaches (pruning, pooling) are compute-light but less expressive; learned resamplers and convolutions are more expressive but add parameters and latency. Adaptive strategies like CrossGET (Shi et al., 2023), VTW (Lin et al., 2025), and MADTP (Cao et al., 2024) leverage cross-modal cues, while VoCo-LLaMA (Ye et al., 2025c) distills compression from LLM attention. Yet, most efforts study compression in isolation, without system-level synergy with speculative decoding. Our work bridges this gap by integrating a question-adaptive, elastic compressor into speculative decoding, compounding gains from fewer visual tokens and fewer target forward passes, while maintaining lossless decoding with respect to the target model.

## 3 METHOD

### 3.1 PRELIMINARIES: BOTTLENECK ANALYSIS

We begin by analyzing the end-to-end latency of speculative decoding (SD) for generating a sequence of length $S$, focusing on the single-batch scenario. In SD, generation proceeds in $R$ rounds; in each round: (1) the draft model $q$ autoregressively proposes $\gamma$ tokens, (2) the target model $p$ verifies these $\gamma$ tokens in a single forward pass, and (3) speculative sampling (Chen et al., 2023) discards any incorrect tokens. Let $T_p(s)$ and $T_q(s)$ denote the time for a single forward pass of the target model $p$ and draft model $q$ respectively, with $s$ tokens. The total latency of SD is:

$$T_{SD} = R \cdot (\gamma \cdot T_q(1) + T_p(\gamma) + T_{\text{sample}}) \tag{1}$$

where $T_{\text{sample}}$ is the (typically negligible) time for speculative sampling in the verification stage.

The speedup over standard autoregressive decoding ($T_{AR}$) (Huang et al., 2025b) is:

$$\text{Speedup} = \frac{T_{AR}}{T_{SD}} = \frac{S \cdot T_p(1)}{R \cdot (\gamma \cdot T_q(1) + T_p(\gamma) + T_{\text{sample}})} = \frac{S}{R} \cdot \frac{1}{\gamma \cdot \frac{T_q(1)}{T_p(1)} + \frac{T_p(\gamma)}{T_p(1)} + \frac{T_{\text{sample}}}{T_p(1)}} \tag{2}$$

Here, we define $\sigma \triangleq \frac{S}{R}$ as the *average accepted tokens per SD round*. Under a simplifying independence assumption with a per-token acceptance probability $\alpha$, a common approximation is $\sigma \approx \sum_{i=1}^{\gamma} \alpha^i = \frac{\alpha(1-\alpha^{\gamma})}{1-\alpha}$. The denominator in equation 2 has three terms: (i) the relative latency of the draft model ($\frac{T_q(1)}{T_p(1)}$), (ii) the cost of multi-token verification ($\frac{T_p(\gamma)}{T_p(1)}$), and (iii) the negligible cost of speculative sampling ($\frac{T_{\text{sample}}}{T_p(1)}$). In practice, speculative decoding tends to be memory-bound (Sadhukhan et al., 2024) at small batch sizes because multi-token verification can saturate memory bandwidth before compute is fully utilized. Hence (ii) and (iii) are usually small; the speedup is most sensitive to the **draft latency** $T_q(1)$ and the **average accepted length** $\sigma$.

### 3.2 CHALLENGES IN SPECULATIVE DECODING FOR VLMS

While speculative decoding is effective for LLMs, its application to VLMs introduces new bottlenecks. In particular, for video (Ji et al., 2025) or long-context inputs (Li et al., 2025b), the draft model's latency is increasingly dominated by the size of the key-value (KV) cache, rather than parameter count alone. As input length grows, the KV cache expands, leading to higher latency—especially in attention layers, where the entire cache must be transferred from high-bandwidth memory (HBM) to on-chip SRAM at each step. Therefore, reducing the draft model's KV cache—**for example via visual token compression**—is essential to sustain speedups in long or high-resolution inputs.

### 3.3 SPECVLM

We first establish EagleVLM as a strong baseline for VLM speculative decoding. EagleVLM integrates a lightweight draft model with the full target VLM, following the EAGLE-2 architecture (Li et al., 2024d). In practice, we observe that compared with the original EAGLE draft model, **the only effective modification is the introduction of an input layer normalization, which stabilizes training and prevents numerical overflow.** The draft processes the same visual and textual inputs as the target but with significantly reduced compute. During inference, the draft proposes multiple tokens, which the target then verifies in parallel. This achieves 1.5–2× speedups over standard autoregressive inference while maintaining output quality. SpecVLM consists of three components: (1) EagleVLM, a strong speculative baseline; (2) an elastic visual compressor that adaptively selects compression strategies; and (3) an online-logit distillation protocol for efficient draft training. Figure 3 overviews the system.

Specifically, **(a) Inference**: Each inference step begins with the Target generating the first token, which serves as the root for the Draft's token tree. Penultimate-layer features from the Target are fused with the Draft's input embeddings autoregressively, enriching representations and mitigating feature mismatch. The same Target features guide the Vision Compressor to select the most suitable operator per question. For simplicity, Top-$k$ branching is omitted in the figure and only a chain is shown. The text embedding and output head layers are shared between Target and Draft. **(b) Training**: SpecVLM *adopts* online distillation of features and logits, unlocking a scaling effect

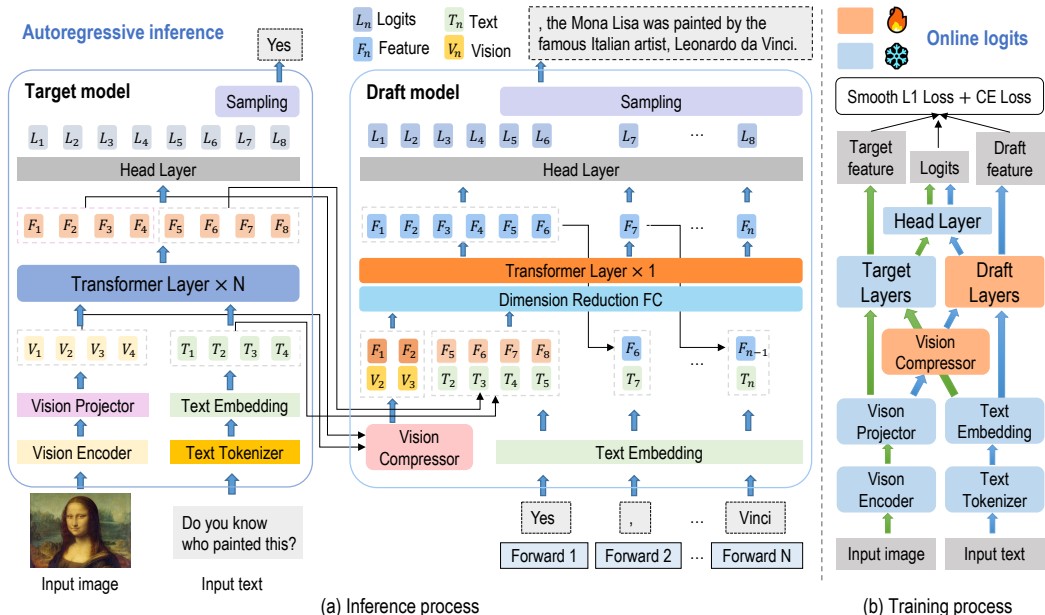

Figure 3: Overview of the SpecVLM architecture. **(a) Inference:** The Target model first generates the initial token, which initializes the Draft model's token tree. Penultimate-layer features from the Target model are fused with the Draft model's input embeddings, and also serve as guidance for the Vision Compressor to adaptively select the optimal compression strategy for each question. The text embedding and output head layers are shared between the Target and Draft models to ensure alignment. **(b) Training:** The Draft model is optimized through joint online distillation of both features and logits, which enables scalable training and reduces overall training cost.

while balancing efficiency and quality. **Furthermore**, the entire training requires only a single end-to-end run, reducing overall cost.

## 3.4 ELASTIC VISION COMPRESSOR

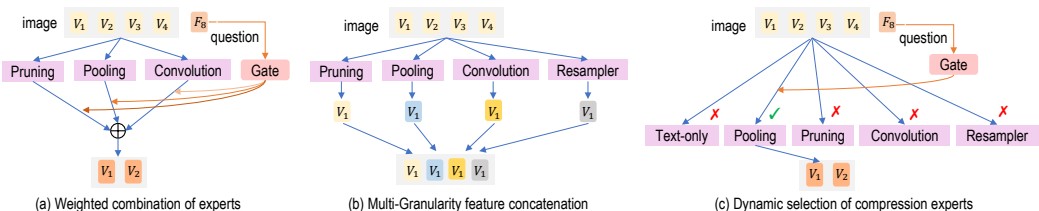

Figure 4: Elastic vision compressor structure. The structure supports three adaptive modes: (a) weighted expert combination via a question-aware gating mechanism, (b) multi-granularity feature concatenation for preserving both global and local information, and (c) dynamic expert selection for resource-efficient inference. This design enables SpecVLM to achieve robust and efficient visual token compression tailored to diverse task requirements and device constraints.

Prior work on visual token compression seldom integrates multiple primitives efficiently. Many pipelines rely on extra clustering (Shang et al., 2024) or multi-stage filtering (Fu et al., 2025). In contrast, we build an elastic compressor using only native operators: **Pruning** drops tokens at random; **Pooling** and **Convolution** are vanilla PyTorch ops; **Resampler** is a simplified Q-Former (Li et al., 2023) with a single cross-attention layer. We find that two object queries provide a good trade-off. This elastic compressor enables adaptive compression under varying task requirements and compute constraints. As illustrated in Figure 4, we instantiate three variants:

**(a) Weighted combination of experts.** Inspired by MoE (Dai et al., 2024), a question-conditioned gate fuses operators with the same compression ratio but different abstraction (Pruning, Pooling, Convolution). This enriches visual features while preserving salient content. Because the convolution branch carries more parameters, extremely high compression is impractical; in practice we cap the ratio at $\leq 5\times$ and use $3\times$ by default for a good accuracy-parameter balance.

**(b) Multi-granularity feature concatenation.** Motivated by dynamic layered sparsity (Yuan et al., 2025), we combine heterogeneous compression scales to retain global context and local details. A configuration that works well is: Pruning/Pooling at $20\times$, Convolution at $3\times$, and Resampler with 2 queries, followed by concatenation.

**(c) Dynamic selection of compression experts.** Evaluating all branches every time is sub-optimal on resource-limited devices. We introduce a question-aware selector that chooses one operator per input based on the gate's top-1 logit, considering question difficulty and image-text relevance. We also include a Text-only branch (drop all vision tokens) to exploit LLMs' zero-shot ability when the question can be answered without visual evidence. This dynamic selection is **the default strategy in SpecVLM** and yields favorable accuracy-compute trade-offs. Since selection is discrete and non-differentiable, we use Gumbel-Softmax (Jang et al., 2016) to approximate the categorical distribution during training; at inference, sampling is replaced by $\mathrm{argmax}$ without extra overhead.

## 3.5 Online-Logit Distillation

Unlike conventional offline distillation that precomputes and stores large volumes of teacher outputs, our approach distills on-the-fly during training, ensuring scalability and up-to-date supervision. At each step, the target (teacher) produces token-level logits $\mathbf{z}_p$ and penultimate-layer features $\mathbf{f}_p$. The draft generates its own logits $\mathbf{z}_q$ and features $\mathbf{f}_q$. We minimize a joint loss that aligns both levels:

$$\mathcal{L}_{\text{online}} = \lambda_{\text{logit}} \, \mathcal{L}_{\text{CE}}(\mathbf{z}_q, \mathbf{z}_p) + \lambda_{\text{feat}} \, \mathcal{L}_{\text{SmoothL1}}(\mathbf{f}_q, \mathbf{f}_p) \tag{3}$$

where $\mathcal{L}_{\text{CE}}$ is cross-entropy between draft and target logits, and $\mathcal{L}_{\text{SmoothL1}}$ is Smooth L1 between their features. Coefficients $\lambda_{\text{logit}}$ (default 0.1) and $\lambda_{\text{feat}}$ (default 1.0) weight logit vs. feature alignment. Empirically, this dual-level supervision improves alignment and increases the average accepted tokens per round, directly boosting end-to-end performance.

## 4 Experiments

### 4.1 Experiment Setup

**Model and Task Setup.** We evaluate SpecVLM on two widely used multimodal benchmarks. The first is the LLaVA benchmark suite (Liu et al., 2023; 2024), which includes LLaVA-Bench-In-the-Wild, MMBench, ScienceQA, SEED-Bench, TextVQA, and VQAv2. For fair comparison and reproducibility, we follow prior work and select the first 60 question-answer pairs containing images from each benchmark. In the original LLaVA, the prompts for MMBench, ScienceQA, SEED-Bench, TextVQA, and VQAv2 were *"Answer with the option's letter from the given choices directly."* This yields very short answers, which under-represents the benefits of speculative decoding. We therefore replace it with: *"Please give a detailed and reasonable explanation for the answer."* The second benchmark is MMMU-V1 (Yue et al., 2024), which spans six disciplines (Art, Biology, Chemistry, Economics, Math, Physics). Because each subject's validation split contains 30 question-answer pairs, we directly use these as the evaluation set. We consider LLaVA-1.5-7B/13B (Liu et al., 2023), LLaVA-1.6-7B/13B (Liu et al., 2024), and Open-LLaVA-1.6-7B (Chen & Xing, 2024) as target models. Both greedy (temperature=0) and stochastic (temperature=1) decoding are evaluated to assess robustness across regimes.

**Metrics.** Consistent with most speculative decoding work, SpecVLM primarily *focuses* on reducing latency rather than maximizing throughput. We use batch size = 1 in all experiments to avoid prompt-length misalignment and padding overhead during speculative sampling. In principle, SpecVLM also *supports* multi-batch inference. We report two key metrics:

- Wall-clock speedup ($\tau$): end-to-end latency reduction compared to standard autoregressive decoding.

- Average accepted tokens ($\sigma$): the average number of tokens accepted by the target per speculative round, reflecting verification efficiency.

As our framework preserves the target distribution (lossless decoding), we do not perform additional output-fidelity evaluation.

**Training and Testing Setup.** We freeze the target VLM and train only the draft model's decoder layer and the vision compressor (Figure 3b). For LLaVA-1.5-7B/13B, we adopt the 655K multi-modal instruction-following dataset from official LLaVA-1.5 visual instruction tuning (150K GPT-generated instructions + 515K academic VQA). For LLaVA-1.6, since the original dataset is un-available, we use the Open-LLaVA-NeXT construction (1.02M samples). We train for 1 epoch with AdamW ($\beta_1{=}0.9$, $\beta_2{=}0.95$), batch size 128, no warmup; learning rate is 8e-5 for 7B and 5e-5 for 13B (weight decay 0). Training runs on $8\times$ AMD MI250 GPUs; one epoch takes 15/20 hours for LLaVA-1.5-7B/13B and 26/35 hours for LLaVA-1.6-7B/13B. We test on AMD Instinct MI250-64G and NVIDIA A100-80G. All latency is measured on a single GPU to ensure consistency across target sizes. Following EAGLE-2, the draft uses tree attention with Top-K=10, total tokens=60, depth=7.

## 4.2 RESULTS

Table 1: Speedup ($\tau$) and average accepted length ($\sigma$) of SpecVLM and EagleVLM on LLaVA benchmarks (epoch=1). Results on MI250-64G. LVA1.5-7B/8B: LLaVA-1.5-7B/13B; LVA1.6-7B/8B: LLaVA-1.6-7B/13B; OLVA1.6-7B: Open-LLaVA-1.6-7B.

| Model | Method | LLaVA-Wild | | MMBench | | ScienceQA | | SEED-Bench | | TextVQA | | VQAv2 | | Avg. | |
|---|---|---|---|---|---|---|---|---|---|---|---|---|---|---|---|
| | | $\tau$ | $\sigma$ | $\tau$ | $\sigma$ | $\tau$ | $\sigma$ | $\tau$ | $\sigma$ | $\tau$ | $\sigma$ | $\tau$ | $\sigma$ | $\tau$ | $\sigma$ |
| | | | | | | | **Temperature = 0** | | | | | | | | |
| LVA1.5-7B | EagleVLM | 2.09 | 3.55 | 1.83 | 3.20 | 1.86 | 3.12 | 2.06 | 3.93 | 1.89 | 3.36 | 2.32 | 4.36 | 2.01 | 3.59 |
| LVA1.5-7B | SpecVLM | **2.20** | **3.68** | **1.93** | **3.29** | **1.93** | **3.19** | **2.21** | **3.99** | **2.01** | **3.49** | **2.39** | **4.39** | **2.11** | **3.67** |
| LVA1.6-7B | EagleVLM | 1.91 | 3.29 | 2.04 | 3.36 | 1.85 | 3.23 | 1.87 | 3.97 | 1.69 | 3.32 | 1.91 | 3.99 | 1.88 | 3.53 |
| LVA1.6-7B | SpecVLM | **2.03** | **3.40** | **2.17** | **3.43** | **1.92** | **3.33** | **2.01** | **4.19** | **1.95** | **3.61** | **2.07** | **4.19** | **2.03** | **3.69** |
| OLVA1.6-7B | EagleVLM | 2.11 | 3.70 | 2.16 | 3.64 | 2.15 | 3.68 | 2.22 | 4.22 | 2.03 | 3.71 | 2.20 | 3.95 | 2.14 | 3.82 |
| OLVA1.6-7B | SpecVLM | **2.18** | **3.74** | **2.41** | **3.82** | **2.35** | **3.86** | **2.27** | **4.33** | **2.15** | **3.77** | **2.27** | **3.97** | **2.27** | **3.91** |
| LVA1.5-13B | EagleVLM | 2.31 | 3.62 | 2.09 | 3.27 | 1.82 | 3.25 | **2.24** | 3.92 | 2.02 | 3.45 | 2.44 | 4.34 | 2.15 | 3.64 |
| LVA1.5-13B | SpecVLM | **2.41** | **3.63** | **2.29** | **3.30** | **1.95** | **3.31** | 2.22 | **4.03** | **2.13** | **3.53** | **2.64** | **4.45** | **2.27** | **3.71** |
| LVA1.6-13B | EagleVLM | 2.29 | 3.82 | 2.54 | 3.85 | 2.39 | 3.76 | 1.91 | 4.49 | 2.14 | 3.90 | 2.29 | 4.70 | 2.26 | 4.09 |
| LVA1.6-13B | SpecVLM | **2.38** | **3.87** | **2.70** | **3.92** | **2.46** | **3.79** | **2.12** | **4.63** | **2.24** | **3.94** | **2.42** | **4.73** | **2.39** | **4.15** |
| | | | | | | | **Temperature = 1** | | | | | | | | |
| LVA1.5-7B | EagleVLM | 1.70 | 2.86 | 1.69 | 2.73 | 1.48 | 2.61 | 1.76 | 3.03 | 1.57 | 2.61 | 1.83 | 3.35 | 1.67 | 2.86 |
| LVA1.5-7B | SpecVLM | **1.77** | **2.92** | **1.84** | **2.74** | **1.62** | **2.68** | **1.84** | **3.12** | **1.63** | **2.73** | **1.98** | **3.43** | **1.78** | **2.94** |
| LVA1.6-7B | EagleVLM | 1.64 | 2.70 | 1.61 | 2.69 | 1.63 | 2.67 | 1.76 | 3.08 | 1.68 | 2.68 | 1.65 | 3.21 | 1.66 | 2.84 |
| LVA1.6-7B | SpecVLM | **1.76** | **2.86** | **1.75** | **2.77** | **1.76** | **2.72** | **1.82** | **3.23** | **1.72** | **2.73** | **1.81** | **3.41** | **1.77** | **2.95** |
| OLVA1.6-7B | EagleVLM | 1.73 | 3.04 | **1.78** | **2.96** | 1.75 | 2.91 | 1.85 | 3.26 | 1.68 | 2.86 | 1.78 | 3.05 | 1.76 | 3.01 |
| OLVA1.6-7B | SpecVLM | **1.82** | **3.15** | 1.77 | 2.88 | **1.76** | **2.93** | **1.89** | **3.28** | **1.69** | **2.88** | **1.81** | **3.09** | **1.79** | **3.03** |
| LVA1.5-13B | EagleVLM | 1.85 | 2.91 | 1.73 | 2.74 | 1.57 | 2.77 | 1.93 | 3.14 | 1.77 | 2.73 | 2.02 | 3.42 | 1.81 | 2.95 |
| LVA1.5-13B | SpecVLM | **1.90** | **2.94** | **1.99** | **2.81** | **1.71** | **2.79** | **2.02** | **3.34** | **1.90** | **2.88** | **2.13** | **3.60** | **1.94** | **3.06** |
| LVA1.6-13B | EagleVLM | 1.90 | 3.11 | 1.95 | 3.01 | 1.99 | 2.99 | 2.10 | 3.57 | 1.88 | 3.18 | 2.13 | 3.65 | 1.99 | 3.25 |
| LVA1.6-13B | SpecVLM | **1.94** | **3.15** | **2.00** | **3.14** | **2.07** | **3.02** | **2.16** | **3.63** | **1.93** | **3.26** | **2.17** | **3.68** | **2.04** | **3.31** |

Table 2: Speedup ($\tau$) and average accepted length ($\sigma$) of SpecVLM and EagleVLM on the official MMMU-V1 benchmarks (T=0, epoch=1). Results are reported on MI250-64G.

| Model | Method | Art | | Biology | | Chemistry | | Economics | | Math | | Physics | | Avg. | |
|---|---|---|---|---|---|---|---|---|---|---|---|---|---|---|---|
| | | $\tau$ | $\sigma$ | $\tau$ | $\sigma$ | $\tau$ | $\sigma$ | $\tau$ | $\sigma$ | $\tau$ | $\sigma$ | $\tau$ | $\sigma$ | $\tau$ | $\sigma$ |
| LVA1.5-7B | EagleVLM | 1.51 | 2.62 | 1.87 | 3.03 | 1.76 | 2.92 | 1.96 | 3.11 | 1.84 | 2.87 | 2.01 | 3.14 | 1.82 | 2.95 |
| LVA1.5-7B | SpecVLM | **1.56** | **2.73** | **1.92** | **3.06** | **1.78** | **2.98** | **2.06** | **3.16** | **2.01** | **3.10** | **2.03** | **3.16** | **1.89** | **3.03** |
| LVA1.6-7B | EagleVLM | 1.83 | 3.29 | 1.87 | 3.11 | 1.80 | 2.93 | 2.13 | 3.41 | 2.12 | 3.41 | 2.01 | 3.22 | 1.96 | 3.23 |
| LVA1.6-7B | SpecVLM | **1.90** | **3.38** | **1.93** | **3.18** | **1.85** | **3.01** | **2.19** | **3.57** | **2.20** | **3.64** | **2.06** | **3.36** | **2.02** | **3.36** |
| OLVA1.6-7B | EagleVLM | 1.86 | 3.30 | 2.00 | 3.43 | 1.98 | 3.31 | 2.13 | 3.55 | 2.46 | 3.95 | 2.12 | 3.55 | 2.09 | 3.52 |
| OLVA1.6-7B | SpecVLM | **2.05** | **3.84** | **2.17** | **3.84** | **2.07** | **3.64** | **2.28** | **3.79** | **2.53** | **4.12** | **2.27** | **3.84** | **2.23** | **3.85** |
| LVA1.5-13B | EagleVLM | 1.65 | 2.80 | 2.07 | 3.24 | 1.89 | 2.92 | 2.19 | 3.32 | 2.09 | 3.02 | 2.14 | 3.19 | 2.01 | 3.08 |
| LVA1.5-13B | SpecVLM | **1.75** | **2.86** | **2.11** | **3.42** | **1.93** | **2.98** | **2.28** | **3.41** | **2.13** | **3.12** | **2.22** | **3.23** | **2.07** | **3.17** |
| LVA1.6-13B | EagleVLM | 2.07 | 3.71 | 2.16 | 3.65 | 2.32 | 3.58 | 2.55 | 3.86 | 2.59 | 3.95 | 2.37 | 3.75 | 2.34 | 3.75 |
| LVA1.6-13B | SpecVLM | **2.14** | **3.86** | **2.26** | **3.74** | **2.36** | **3.64** | **2.62** | **3.97** | **2.63** | **4.02** | **2.44** | **3.86** | **2.41** | **3.85** |

**Analysis.** Tables 1 and 2 compare SpecVLM and EagleVLM across model sizes, benchmarks, and subject domains. SpecVLM consistently improves both speedup ($\tau$) and average accepted length ($\sigma$). For instance, on LLaVA-1.6-13B, SpecVLM reaches $2.38\times$ speedup on LLaVA-Wild and up to

2.70× on MMBench, with $\sigma$>3.8 tokens per round. On MMMU-V1, SpecVLM with LVA1.6-13B attains 2.14× on Art and up to 2.63× on Math, while maintaining high $\sigma$.

A trend across benchmarks is that the SpecVLM-EagleVLM gap widens with model size, indicating stronger gains where computation is the bottleneck. Maintaining higher $\sigma$ directly reduces required target forward passes and improves wall-clock latency. Beyond raw speed, the elastic compressor appears to refine visual features by filtering distractions, yielding higher $\sigma$ than the baseline even at similar compression levels. SpecVLM delivers stable improvements in both deterministic (T=0) and stochastic (T=1) decoding, enabling a favorable balance between efficiency and quality without altering the target distribution.

# 5 ABLATION STUDIES

## 5.1 COMPONENT ANALYSIS

We ablate the contribution of EagleVLM, the elastic visual compressor, and online-logit distillation on LLaVA-1.6-7B (Table 3). Configurations include: (1) **EagleVLM** (EAGLE-2-style speculative baseline); (2) **Weight** (weighted fusion of compression experts); (3) **Concat** (multi-granularity feature concatenation); and (4) **Select** (dynamic expert selection; default SpecVLM). The results show cumulative gains. Starting from EagleVLM, adding weighted fusion improves both $\tau$ and $\sigma$. Multi-granularity concatenation preserves these gains. Dynamic selection (full SpecVLM) yields the best overall performance, reaching 2.03× average speedup and 3.69 average accepted length. These findings highlight the importance of adaptability—particularly the question-aware expert selection—in achieving robust speed-accuracy trade-offs.

Table 3: Ablation study of key components on LLaVA-1.6-7B across LLaVA benchmarks (T=0, epoch=1). Results are reported on MI250-64G. EagleVLM: EAGLE-2-style speculative decoding baseline; Weight: Weighted combination of experts; Concat: Multi-granularity feature concatenation; Select: Dynamic selection of compression experts.

| Configuration | LLaVA-Wild | | MMBench | | ScienceQA | | SEED-Bench | | TextVQA | | VQAv2 | | Avg. | |
|---|---|---|---|---|---|---|---|---|---|---|---|---|---|---|
| | $\tau$ | $\sigma$ | $\tau$ | $\sigma$ | $\tau$ | $\sigma$ | $\tau$ | $\sigma$ | $\tau$ | $\sigma$ | $\tau$ | $\sigma$ | $\tau$ | $\sigma$ |
| EagleVLM | 1.91 | 3.29 | 2.04 | 3.36 | 1.85 | 3.23 | 1.87 | 3.97 | 1.69 | 3.32 | 1.91 | 3.99 | 1.88 | 3.53 |
| w/ Weight | 1.98 | 3.36 | 2.12 | 3.41 | 1.90 | 3.38 | 1.99 | 3.61 | 1.93 | 3.59 | 2.01 | 4.29 | 1.99 | 3.66 |
| w/ Concat | 1.95 | 3.32 | 2.08 | 3.39 | 1.88 | 3.51 | 1.96 | 3.58 | 1.91 | 3.43 | 1.96 | 4.18 | 1.96 | 3.57 |
| w/ Select | **2.03** | **3.40** | **2.17** | **3.43** | **1.92** | **3.33** | **2.01** | **4.19** | **1.95** | **3.61** | **2.07** | **4.19** | **2.03** | **3.69** |

Table 4: Isolated branches within the elastic vision compressor of LLaVA-1.6-7B on the LLaVA-Bench-In-the-Wild (T=0, epoch=1). Each branch is tested with specified compression settings while others remain constant. Wall-clock speedup ($\tau$) and average accepted length ($\sigma$) on MI250-64G.

| Branch | Ratio | $\tau$ | $\sigma$ | Branch | Ratio | $\tau$ | $\sigma$ | Branch | Ratio | $\tau$ | $\sigma$ | Branch | num | $\tau$ | $\sigma$ |
|---|---|---|---|---|---|---|---|---|---|---|---|---|---|---|---|
| Prun | 2× | 1.95 | 3.35 | Pool | 2× | 1.97 | 3.36 | Conv | 2× | 1.91 | 3.29 | Resamp | 1 | 1.92 | 3.30 |
| Prun | 3× | 1.98 | 3.37 | Pool | 3× | 1.99 | 3.39 | Conv | 3× | **2.03** | **3.40** | Resamp | 2 | **2.03** | **3.40** |
| Prun | 5× | 2.00 | 3.39 | Pool | 5× | 2.01 | 3.37 | Conv | 5× | 1.98 | 3.35 | Resamp | 4 | 1.99 | 3.36 |
| Prun | 10× | 1.99 | 3.38 | Pool | 10× | 1.98 | 3.38 | Conv | 10× | 1.91 | 3.29 | Resamp | 8 | 2.01 | 3.39 |
| Prun | 20× | **2.03** | **3.40** | Pool | 20× | **2.03** | **3.40** | Conv | 20× | 1.88 | 2.90 | Resamp | 10 | 1.98 | 3.34 |
| Prun | 30× | 2.00 | 3.36 | Pool | 30× | 2.01 | 3.37 | Conv | 30× | 1.67 | 2.73 | Resamp | 20 | 1.99 | 3.36 |

**Analysis of Isolated Branches.** From Table 4, we observe that pruning and pooling, which are parameter-free, maintain robustness up to a 10-20× compression ratio, improving $\tau$ while keeping $\sigma$ high. The resampler performs best with 2 queries, as more queries add overhead without boosting $\sigma$. Lightweight convolutions match pooling at 3-5× ratios, but both $\tau$ and $\sigma$ drop at ratios of 10× or more. These insights guide our strategy: use 10-20× pooling/pruning for simple or text-based tasks, 3× convolution and a 2-query resampler for detailed visual information inputs.

## 5.2 TRAINING-TIME SCALING

Figures 5 and 6 illustrate that both accepted length ($\sigma$) and speedup ($\tau$) consistently rise with each epoch across all variants and sizes. This suggests that prolonged online training enhances draft-

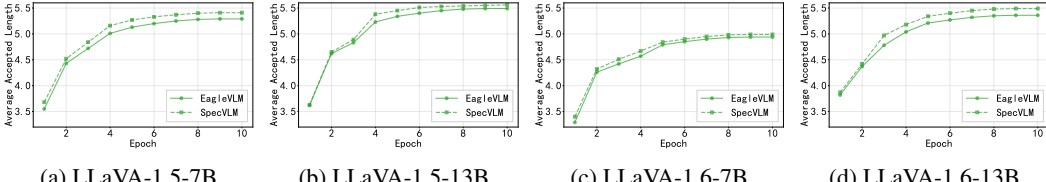

(a) LLaVA-1.5-7B      (b) LLaVA-1.5-13B      (c) LLaVA-1.6-7B      (d) LLaVA-1.6-13B

Figure 5: Training-time scaling on LLaVA-Bench-In-the-Wild (T=0). Average accepted length ($\sigma$) is reported on MI250-64G. Across all variants, $\sigma$ increases as training progresses.

target alignment, boosts accepted tokens per round, and increases acceleration. Notably, these results underscore the potential of online-logit distillation in improving VLM speculative decoding, without requiring architectural changes or increased draft capacity. This effect is consistently observed across various model sizes, demonstrating strong scalability. By the 5th epoch, the draft model starts to converge, achieving optimal performance by the 8th epoch. However, further extending the training period would substantially raise computational costs, so we selected the 5th epoch as the optimal trade-off point.

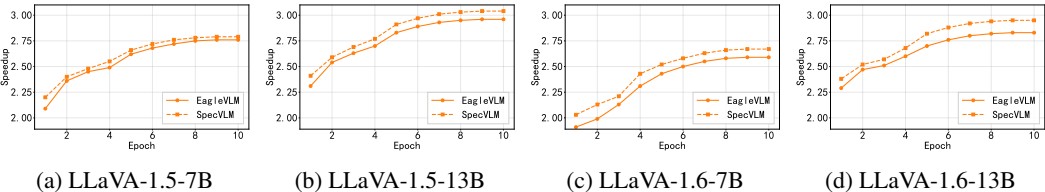

(a) LLaVA-1.5-7B      (b) LLaVA-1.5-13B      (c) LLaVA-1.6-7B      (d) LLaVA-1.6-13B

Figure 6: Training-time scaling on LLaVA-Bench-In-the-Wild (T=0). Speedup ($\tau$) is reported on MI250-64G. Across all variants, $\tau$ improves with additional epochs.

## 6 DISCUSSION AND LIMITATIONS

Although our current SpecVLM represents an initial exploration of speculative decoding for VLMs with promising results, several limitations remain. First, the vision compressor's compression scales are still manually configured; an instance-wise mechanism to adjust compression ratios is a valuable direction. Second, exploring dynamic compression of the draft's KV cache at inference time could further improve efficiency. Additionally, while we focus on training time, other scaling axes—larger training corpora and deeper drafts—may also improve speculative efficiency. These dimensions could reveal broader scaling laws for speculative decoding in VLMs. Finally, we do not exhaustively tune draft hyperparameters (e.g., depth, tree Top-K, number of nodes) or optimization settings (e.g., learning rate, batch size). Reported performance may therefore be suboptimal; we hope future work will build on our codebase to identify best settings.

## 7 CONCLUSION

In this work, we introduce SpecVLM, a practical and scalable framework for accelerating vision-language models (VLMs) through speculative decoding and elastic visual token compression. Building on a strong EAGLE-2-style speculative baseline (EagleVLM), SpecVLM incorporates two core innovations: an elastic visual compressor that adaptively selects among multiple compression primitives, and an online logit distillation protocol that enables efficient draft model training without reliance on large offline datasets. Extensive experiments on LLaVA benchmarks and MMMU evaluation suites demonstrate that SpecVLM consistently delivers superior speedup and efficiency gains across diverse model sizes and task complexities, all while preserving output quality. Notably, we observe a robust training-time scaling effect, where simply extending the duration of online draft training yields substantial improvements in both average accepted length and inference speedup, independent of model capacity or dataset size. This highlights the potential for further efficiency gains through targeted training strategies.

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

## A APPENDIX

### A.1 ADDITIONAL EXPERIMENTAL RESULTS.

**Results on A100-80G:** The following results are obtained from testing on an NVIDIA A100-80G.

Table 5: Speedup ($\tau$) and average accepted length ($\sigma$) of SpecVLM and EagleVLM on LLaVA benchmarks (epoch=1). Results on A100-80G. LVA1.5-7B/8B: LLaVA-1.5-7B/13B; LVA1.6-7B/8B: LLaVA-1.6-7B/13B; OLVA1.6-7B: Open-LLaVA-1.6-7B.

| Model | Method | LLaVA-Wild | | MMBench | | ScienceQA | | SEED-Bench | | TextVQA | | VQAv2 | | Avg. | |
|---|---|---|---|---|---|---|---|---|---|---|---|---|---|---|---|
| | | $\tau$ | $\sigma$ | $\tau$ | $\sigma$ | $\tau$ | $\sigma$ | $\tau$ | $\sigma$ | $\tau$ | $\sigma$ | $\tau$ | $\sigma$ | $\tau$ | $\sigma$ |
| | | | | | | | Temperature=0 | | | | | | | | |
| LVA1.5-7B | EagleVLM | 1.91 | 3.57 | 1.70 | 3.21 | 1.64 | 3.09 | 1.93 | 3.92 | 1.64 | 3.35 | 2.07 | 4.16 | 1.82 | 3.55 |
| LVA1.5-7B | SpecVLM | **2.14** | **3.69** | **1.89** | **3.30** | **1.70** | **3.16** | **1.95** | **4.04** | **1.77** | **3.49** | **2.15** | **4.22** | **1.93** | **3.65** |
| LVA1.6-7B | EagleVLM | 1.59 | 3.27 | 1.92 | 3.39 | 1.65 | 3.26 | 1.61 | 4.00 | 1.50 | 3.39 | 1.62 | 3.92 | 1.65 | 3.54 |
| LVA1.6-7B | SpecVLM | **1.79** | **3.40** | 1.91 | **3.44** | **1.72** | **3.33** | **1.71** | **4.20** | **1.66** | **3.53** | **1.83** | **4.15** | **1.77** | **3.68** |
| OLVA1.6-7B | EagleVLM | 1.86 | 3.69 | 1.70 | 3.63 | 1.84 | 3.70 | 1.80 | 4.21 | 1.67 | 3.69 | 1.88 | 3.94 | 1.79 | 3.81 |
| OLVA1.6-7B | SpecVLM | **2.02** | **3.75** | **1.79** | **3.82** | **2.01** | **3.88** | **1.99** | **4.36** | **1.79** | **3.76** | **2.01** | **3.95** | **1.93** | **3.92** |
| LVA1.5-13B | EagleVLM | 1.95 | 3.60 | 1.76 | 3.22 | 1.72 | 3.19 | 1.94 | 3.89 | 1.84 | 3.45 | 2.35 | 4.34 | 1.93 | 3.61 |
| LVA1.5-13B | SpecVLM | **1.99** | **3.63** | **1.91** | **3.24** | **1.80** | **3.26** | **2.04** | **3.98** | **1.89** | **3.52** | **2.48** | **4.44** | **2.02** | **3.68** |
| LVA1.6-13B | EagleVLM | 1.85 | 3.77 | 2.20 | 3.78 | 2.03 | 3.74 | 1.66 | 4.52 | 1.65 | 3.87 | 1.98 | 4.69 | 1.90 | 4.06 |
| LVA1.6-13B | SpecVLM | **1.94** | **3.82** | **2.28** | **3.86** | **2.09** | **3.79** | **1.76** | **4.67** | **1.72** | **3.91** | **2.03** | **4.72** | **1.97** | **4.13** |
| | | | | | | | Temperature=1 | | | | | | | | |
| LVA1.5-7B | EagleVLM | 1.49 | 2.83 | 1.57 | 2.72 | 1.45 | 2.63 | 1.64 | 3.02 | 1.43 | 2.63 | 1.81 | 3.32 | 1.56 | 2.86 |
| LVA1.5-7B | SpecVLM | **1.61** | **2.89** | **1.67** | **2.73** | **1.53** | **2.74** | **1.68** | **3.12** | **1.47** | **2.73** | 1.81 | **3.47** | **1.63** | **2.95** |
| LVA1.6-7B | EagleVLM | 1.41 | 2.69 | 1.48 | 2.72 | 1.42 | 2.71 | 1.36 | 3.10 | 1.46 | 2.67 | 1.67 | 3.15 | 1.47 | 2.84 |
| LVA1.6-7B | SpecVLM | **1.55** | **2.80** | **1.60** | **2.80** | **1.47** | **2.83** | **1.55** | **3.24** | **1.52** | **2.79** | **1.77** | **3.25** | **1.57** | **2.95** |
| OLVA1.6-7B | EagleVLM | 1.44 | 3.02 | 1.51 | 2.92 | 1.46 | 2.87 | 1.57 | 3.25 | 1.37 | 2.86 | 1.75 | 3.21 | 1.52 | 3.02 |
| OLVA1.6-7B | SpecVLM | **1.52** | **3.14** | **1.56** | **2.99** | **1.50** | **2.96** | **1.64** | **3.46** | **1.40** | **2.88** | **1.80** | **3.33** | **1.57** | **3.13** |
| LVA1.5-13B | EagleVLM | 1.83 | 2.94 | 1.69 | 2.83 | 1.36 | 2.77 | 1.68 | 3.14 | 1.66 | 2.86 | 1.75 | 3.47 | 1.66 | 3.00 |
| LVA1.5-13B | SpecVLM | 1.81 | **2.99** | **1.80** | **2.86** | **1.44** | **2.78** | **1.85** | **3.25** | **1.67** | **2.97** | **1.76** | **3.60** | **1.72** | **3.07** |
| LVA1.6-13B | EagleVLM | 1.68 | 3.13 | 1.76 | 3.11 | 1.62 | 3.11 | 1.88 | 3.53 | 1.55 | 3.11 | 1.78 | 3.70 | 1.71 | 3.28 |
| LVA1.6-13B | SpecVLM | **1.78** | **3.24** | **1.85** | **3.26** | **1.68** | **3.29** | **2.03** | **3.67** | **1.58** | **3.13** | **1.86** | **3.78** | **1.80** | **3.40** |

**Analysis:** Table 5 presents a comprehensive comparison between SpecVLM and EagleVLM across multiple LLaVA benchmarks and model variants. Across all datasets and model sizes, SpecVLM consistently outperforms EagleVLM in both speedup ($\tau$) and average accepted length ($\sigma$) under deterministic (T=0) and stochastic (T=1) decoding. Improvements are more pronounced for larger models and harder benchmarks, indicating that speculative decoding with adaptive visual compression scales with capacity and task difficulty. These results show higher efficiency without compromising output quality, as reflected by increased $\sigma$, and validate the robustness of SpecVLM across diverse VLM architectures and evaluation scenarios.

Table 6: Speedup ($\tau$) and average accepted length ($\sigma$) of SpecVLM and EagleVLM on the official MMMU-V1 benchmarks (T=0, epoch=1). Results are reported on A100-80G.

| Model | Method | Art | | Biology | | Chemistry | | Economics | | Math | | Physics | | Avg. | |
|---|---|---|---|---|---|---|---|---|---|---|---|---|---|---|---|
| | | $\tau$ | $\sigma$ | $\tau$ | $\sigma$ | $\tau$ | $\sigma$ | $\tau$ | $\sigma$ | $\tau$ | $\sigma$ | $\tau$ | $\sigma$ | $\tau$ | $\sigma$ |
| LVA1.5-7B | EagleVLM | 1.32 | 2.57 | 1.61 | 3.11 | 1.61 | 3.07 | 1.70 | 3.17 | 1.90 | 3.42 | 1.74 | 3.13 | 1.65 | 3.08 |
| LVA1.5-7B | SpecVLM | **1.41** | **2.65** | **1.66** | **3.18** | **1.65** | **3.09** | **1.83** | **3.25** | **1.94** | **3.53** | **1.82** | **3.23** | **1.72** | **3.16** |
| LVA1.6-7B | EagleVLM | 1.52 | 3.37 | 1.51 | 3.12 | 1.59 | 3.09 | 1.79 | 3.40 | 1.78 | 3.39 | 1.73 | 3.24 | 1.65 | 3.27 |
| LVA1.6-7B | SpecVLM | **1.59** | **3.49** | **1.54** | **3.21** | **1.60** | **3.15** | **1.81** | **3.46** | **1.86** | **3.48** | **1.84** | **3.36** | **1.71** | **3.36** |
| OLVA1.6-7B | EagleVLM | 1.61 | 3.27 | 1.73 | 3.48 | 1.61 | 3.21 | 1.76 | 3.51 | 1.85 | 3.70 | 1.77 | 3.52 | 1.72 | 3.45 |
| OLVA1.6-7B | SpecVLM | **1.82** | **3.71** | **1.92** | **3.82** | **1.73** | **3.57** | **1.93** | **3.85** | **1.93** | **3.83** | **1.97** | **3.68** | **1.88** | **3.74** |
| LVA1.5-13B | EagleVLM | 1.57 | 2.78 | 1.82 | 3.23 | 1.66 | 2.87 | 1.86 | 3.28 | 1.93 | 3.27 | 1.82 | 3.21 | 1.77 | 3.11 |
| LVA1.5-13B | SpecVLM | **1.64** | **2.92** | **1.86** | **3.27** | **1.74** | **2.93** | **1.91** | **3.34** | **2.00** | **3.31** | **1.87** | **3.26** | **1.84** | **3.17** |
| LVA1.6-13B | EagleVLM | 1.75 | 3.70 | 1.86 | 3.63 | 2.08 | 3.63 | 2.14 | 3.80 | 2.12 | 3.87 | 1.97 | 3.65 | 1.99 | 3.71 |
| LVA1.6-13B | SpecVLM | **1.76** | **3.75** | **1.90** | **3.69** | **2.14** | **3.76** | **2.19** | **3.83** | **2.18** | **3.92** | **2.02** | **3.78** | **2.03** | **3.79** |

**Analysis:** Table 6 reports SpecVLM vs. EagleVLM on MMMU-V1 across academic subjects. SpecVLM consistently achieves higher speedups and longer accepted lengths across all subjects and model variants. Gains are especially notable in Math and Physics, highlighting robust acceleration and efficient speculative verification for complex reasoning. These results confirm the effectiveness and scalability of our methods across evaluation settings and subject areas.

## A.2 DETAILED TRAINING-TIME SCALING

Table 7: Training-time scaling on LLaVA-Bench-In-the-Wild (T=0). Speedup ($\tau$) and average accepted length ($\sigma$) are reported on MI250-64G.

| Model | Epoch | EagleVLM | | SpecVLM | |
|---|---|---|---|---|---|
| | | $\tau$ | $\sigma$ | $\tau$ | $\sigma$ |
| LLaVA-1.5-7B | 1 | 2.09 | 3.55 | 2.20 | 3.68 |
| LLaVA-1.5-7B | 2 | 2.36 | 4.43 | 2.39 | 4.52 |
| LLaVA-1.5-7B | 3 | 2.45 | 4.72 | 2.47 | 4.84 |
| LLaVA-1.5-7B | 4 | 2.49 | 5.01 | 2.55 | 5.16 |
| LLaVA-1.5-7B | 5 | 2.62 | 5.13 | 2.66 | 5.27 |
| LLaVA-1.6-7B | 1 | 1.91 | 3.29 | 2.03 | 3.40 |
| LLaVA-1.6-7B | 2 | 1.99 | 4.26 | 2.03 | 4.32 |
| LLaVA-1.6-7B | 3 | 2.13 | 4.42 | 2.21 | 4.51 |
| LLaVA-1.6-7B | 4 | 2.31 | 4.57 | 2.33 | 4.67 |
| LLaVA-1.6-7B | 5 | 2.43 | 4.79 | 2.52 | 4.84 |
| Open-LLaVA-1.6-7B | 1 | 2.11 | 3.70 | 2.18 | 3.74 |
| Open-LLaVA-1.6-7B | 2 | 2.18 | 4.31 | 2.22 | 4.42 |
| Open-LLaVA-1.6-7B | 3 | 2.30 | 4.53 | 2.35 | 4.60 |
| Open-LLaVA-1.6-7B | 4 | 2.38 | 4.71 | 2.46 | 4.88 |
| Open-LLaVA-1.6-7B | 5 | 2.47 | 4.96 | 2.53 | 5.05 |
| LLaVA-1.5-13B | 1 | 2.31 | 3.62 | 2.41 | 3.63 |
| LLaVA-1.5-13B | 2 | 2.54 | 4.62 | 2.56 | 4.65 |
| LLaVA-1.5-13B | 3 | 2.63 | 4.83 | 2.69 | 4.89 |
| LLaVA-1.5-13B | 4 | 2.70 | 5.23 | 2.77 | 5.38 |
| LLaVA-1.5-13B | 5 | 2.83 | 5.34 | 2.91 | 5.45 |
| LLaVA-1.6-13B | 1 | 2.29 | 3.82 | 2.38 | 3.87 |
| LLaVA-1.6-13B | 2 | 2.47 | 4.37 | 2.51 | 4.42 |
| LLaVA-1.6-13B | 3 | 2.51 | 4.78 | 2.55 | 4.97 |
| LLaVA-1.6-13B | 4 | 2.60 | 5.04 | 2.68 | 5.18 |
| LLaVA-1.6-13B | 5 | 2.70 | 5.21 | 2.82 | 5.34 |

**Analysis:** Table 7 shows that as epochs increase, both $\tau$ and $\sigma$ rise for all models. SpecVLM maintains a consistent advantage over EagleVLM at every epoch, with the gap widening over time. This indicates that longer online training enhances speculative efficiency and compounds the benefits of the elastic compressor. Sufficient training is thus important to realize SpecVLM's gains across sizes and regimes.

**Training-time Scaling for Open-LLaVA-1.6-7B:**

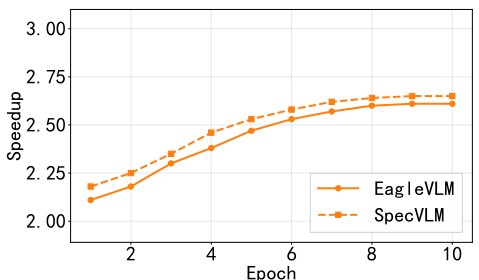
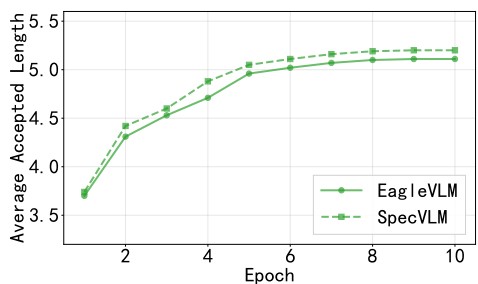

(a) Speedup ($\tau$) across epochs.    (b) Average accepted length ($\sigma$) across epochs.

Figure 7: Training-time scaling of Open-LLaVA-1.6-7B on LLaVA-Bench-In-the-Wild (T=0). Results are reported on MI250-64G. The left subfigure shows speedup ($\tau$) as training progresses; the right shows average accepted length ($\sigma$). Both metrics increase with training epochs, underscoring scalability.

## A.3 THE USE OF LARGE LANGUAGE MODELS (LLMS)

This work only used LLMs for language polishing and did not involve any technical issues related to the work.

