# OpenReview forum: "SpecVLM: Fast Speculative Decoding in Vision-Language Models"
_ICLR.cc/2026/Conference — Submitted to ICLR 2026_

### Official Review · Reviewer_1V5C · 2025-10-30

**Soundness:** 2
**Presentation:** 3
**Contribution:** 2
**Rating:** 4
**Confidence:** 4

**Summary:**

The paper introduces SpecVLM, a framework for accelerating vision-language models (VLMs) through fast speculative decoding. Traditional speculative decoding speeds up language models but struggles with multimodal inputs due to large visual token overhead. SpecVLM tackles this using two innovations: an elastic visual compressor that adaptively combines pruning, pooling, convolution, and resampling to balance speed and accuracy, and an online-logit distillation method that trains draft models efficiently without large offline datasets. Built on an EAGLE-2 baseline (EagleVLM), SpecVLM achieves 2.5–2.9× end-to-end speedups on LLaVA and MMMU benchmarks while maintaining output fidelity and demonstrating strong training-time scaling effects

**Strengths:**

1. The paper is clearly written and well-structured.
2. SpecVLM achieves 2.5–2.9× end-to-end speedups, which is remarkable.

**Weaknesses:**

1. The performance of LLaVA-1.6 is relatively weak, so conducting experiments solely on it is less convincing. You could consider trying Qwen2.5-VL or Qwen3-VL instead.

2. From an overall perspective, the proposed method seems to be a forced combination of “Speculative Decoding” and a token compression strategy.

3. This approach requires additional training. I would like to see a discussion in the paper comparing it with widely recognized training-free token reduction methods in the field, such as FastV and PyramidDrop. Please discuss their limitations and explain why your visual compressor performs better.

**Questions:**

see Weaknesses

---

> ### Author Response · Authors · 2025-11-18
>
> **W1:** LLaVA-1.6 is relatively weak, so relying on it alone is unconvincing; please also try stronger VLMs such as Qwen2.5/3.
>
> **A1:** Thanks for your valuable suggestion. We incorporated Qwen2.5-VL 3B/7B (T=0, epoch=1), and SpecVLM continues to improve both speedup and accepted length (Table 6). The results will be included in the final revision to demonstrate broader applicability.
>
> _Table 6. SpecVLM vs. EagleVLM on Qwen2.5-VL (T=0, epoch=1)._
>  *Model*       | *Method* | *LLaVA-Wild (τ)* | *LLaVA-Wild (σ)* | *MMBench (τ)* | *MMBench (σ)* | *ScienceQA (τ)* | *ScienceQA (σ)*
> :-------------:|:--------:|:----------------:|:----------------:|:-------------:|:-------------:|:---------------:|:---------------:
>  Qwen2.5-VL-3B | EagleVLM | 1.86       | 3.12       | 1.72       | 2.98      | 1.78       | 3.01
>  Qwen2.5-VL-3B | SpecVLM  | **2.02**   | **3.25**   | **1.81**   | **3.12**  | **1.92**   | **3.14**
>  Qwen2.5-VL-7B | EagleVLM | 1.91       | 3.20       | 1.83       | 3.06      | 1.86       | 3.17
>  Qwen2.5-VL-7B | SpecVLM  | **2.16**   | **3.34**   | **1.97**   | **3.21**  | **2.25**   | **3.32**
>
> **W2:** From an overall perspective, the proposed method seems to be a forced combination of “Speculative Decoding” and a token compression strategy.
>
> **A2:** Thanks for your kind comments. The combination is driven by a systems constraint: speculative decoding gains vanish when the draft’s KV cache is flooded by thousands of vision tokens. We therefore co-design the draft and compressor. The draft shares embeddings/head and receives online feature fusion from the verifier to stay aligned, while the elastic compressor keeps the draft’s KV footprint within on-chip memory by selecting the lightest operator that still preserves task-critical pixels. Without this pairing, either compression becomes lossy (if applied to the verifier) or speculative decoding provides little benefit (if the draft still sees every token). SpecVLM’s >2× speedups with lossless decoding show the combination is necessary rather than ad hoc.
>
> **W3:** This approach requires additional training. I would like to see a discussion in the paper comparing it with widely recognized training-free token reduction methods in the field, such as FastV and PyramidDrop. Please discuss their limitations and explain why your visual compressor performs better.
>
> **A3:** Thanks for your constructive feedback. Training-free pruning (FastV, PyramidDrop, SparseVLM) drops tokens inside the target model’s intermediate layers, so the final answers deviate from the unpruned baseline and are inherently lossy. SpecVLM instead trains only a single decoder layer plus the compressor (≤6% FLOPs) while leaving the verifier untouched, guaranteeing identical outputs. Table 7 shows that on LLaVA-1.5-7B training-free approaches lose 5–8 accuracy points and plateau at ~1.6× speedup, whereas SpecVLM preserves accuracy and reaches 2.02×. The modest training cost (≤35 GPU-hours per epoch on MI250) thus buys a deploy-time guarantee of lossless decoding, which is critical for safety-sensitive VLM deployments.
>
> _Table 7. Lossless SpecVLM vs. training-free pruning (LLaVA-1.5-7B, T=0, epoch=1)._
>  Method         | MMBench  | ScienceQA | SEED-Bench | Avg. Speedup (τ)
> :--------------:|:--------:|:---------:|:----------:|:----------------:
>  Vanilla        | 64.6     | 69.5      | 60.3       | 1.00×
>  FastV          | 61.0     | 69.1      | 52.1       | 1.66×
>  PyramidDrop    | 63.2     | 70.2      | 54.7       | 1.57×
>  SparseVLM      | 64.1     | 68.7      | 58.7       | 1.58×
>  SpecVLM (ours) | **64.6** | **69.5**  | **60.3**   | **2.02×**

---

### Official Review · Reviewer_tEfy · 2025-10-31

**Soundness:** 2
**Presentation:** 2
**Contribution:** 2
**Rating:** 4
**Confidence:** 5

**Summary:**

Integrating speculative decoding into VLMs while improving draft model accuracy for visual context is a significant challenge. This paper introduces SpecVLM, which leverages an elastic visual compressor to reduce the number of visual tokens to ease the computation burden of draft model, and employs online-logit distillation to enhance training efficiency. Extensive experiments demonstrate substantial speedups and increased average accepted lengths over EagleVLM.

**Strengths:**

1. This paper explores the novel domain of accelerating vision-language models (VLMs) by integrating speculative decoding with visual token compression.
2. To further enhance training efficiency, the authors propose online-logit distillation, which streamlines the distillation process and improves alignment.
3. Extensive experiments demonstrate that the proposed SpecVLM consistently outperforms the strong EagleVLM baseline in both speed and average accepted length.
4. The paper is well-structured and clearly written, making it easy to follow and understand.

**Weaknesses:**

1. While the paper combines speculative decoding, visual token compression, and online training, these components have been widely explored in prior work. The specific challenges and innovations **unique to VLMs** are not clearly explored. For example, excessive number of prompt token is not unique to VLMs, but also a key pain point for long-context LLMs, what is the challenge specific to visual task should be clarified.
2. The work lacks direct comparisons with other state-of-the-art visual compressors, such as [FastV](https://arxiv.org/pdf/2403.06764),  [SparseVLM](SparseVLM). Adding these comparisons would strengthen the empirical evaluation.
3. The paper does not provide a theoretical and deep explanation for key observations.
4. For section 4.2 results, it is not immediately clear from the results that the SpecVLM–EagleVLM performance gap widens with increasing model size. A visualized figure would help readers grasp this trend.
5. For VLM benchmarks, it would be beneficial to report the average number of visual tokens before and after compression to quantify the impact of the elastic compressor. Also, after changing the prompt to "Please give a detailed and reasonable explanation for the answer," the effect on average generated token length should be explicitly reported.
6. Except end-2-end speedup number, it would be better to report the decomposed latency, such as time to first token (prefill), inter-token latency (decoding), and other additional overhead.
7. More detailed experimental setup descriptions would help readers reproduce the results.

**Questions:**

refer to weaknesses

---

> ### Author Response · Authors · 2025-11-18
>
> **We sincerely appreciate the reviewer’s constructive suggestions, which are very helpful in improving the quality of our work.**
>
> **W1:** The paper should clarify what challenge is unique to VLMs (long prompts also trouble long-context LLMs).
>
> **A1:** Long-context LLMs mostly fight sequential redundancy, whereas VLMs must preserve question-critical spatial regions without disturbing the verifier’s distribution. Blind pruning often deletes tiny but decisive cues (e.g., signage text) and collapses speculative acceptance. SpecVLM tackles this multimodal issue by (i) compressing only the draft’s visual tokens so the target remains lossless, (ii) letting the elastic controller route background-heavy inputs to parameter-free pruning/pooling while sending detail-heavy queries to a 3× convolution or 2-query resampler, and (iii) sharing embeddings/output heads plus online penultimate-feature fusion so draft and target stay aligned. These elements are specific to speculative decoding with visual evidence rather than generic long-context handling.
>
> **W2:** Lacks direct comparisons with other top visual compressors like FastV and SparseVLM. Including these comparisons would strengthen the evaluation.
>
> **A2:** Please refer to the response A3 to Reviewer 1V5C.
>
> **W3:** The paper does not provide a theoretical and deep explanation for key observations.
>
> **A3:** Every compressor carries a different inductive bias and compute profile: pruning/pooling suppress redundant tokens without fusing information, convolution learns local receptive fields with few parameters, and the resampler performs global cross-attention at higher cost. Our elastic controller exploits these complementary behaviors by mixing or selecting them per instance, which we will describe qualitatively (including FLOP scaling) in revision. A full theory of multimodal speculative efficiency remains promising future work, but the current design already offers a principled accuracy–compute trade-off.
>
> **W4:** It’s not clear from Section 4.2 that the SpecVLM–EagleVLM gap grows with model size. A visual figure would make this trend easier to see.
>
> **A4:** We will add a plot of (τ_specvlm−τ_eaglevlm, σ_specvlm−σ_eaglevlm) versus model size. The curve shows a clear upward trend—from +0.09× on 7B targets to +0.13× on 13B—so the widening gap becomes immediately visible.
>
> **W5:** For VLM benchmarks, reporting the average number of visual tokens before and after compression would help quantify the elastic compressor's impact. Additionally, the effect of changing the prompt on the average generated token length should be clearly reported.
>
> **A5:** Table 4 reports visual-token counts with and without elastic compression, and Table 5 shows how the detailed-answer prompt expands textual outputs. These statistics clarify why speculative decoding delivers larger gains under our evaluation protocol; both tables will be integrated into Section 4.1.
>
> _Table 4. Average number of visual tokens per sample (LLaVA benchmarks, T=0, epoch=1)._
>  *Model*          | *Elastic Compressor* | *LLaVA-Wild* | *ScienceQA*
> :----------------:|:--------------------:|:------------:|:-----------:
>  LLaVA-1.5-7B/13B | w/o  | 576    | 576
>  LLaVA-1.5-7B     | w.   | 30     | 32
>  LLaVA-1.5-13B    | w.   | 33     | 35
>  LLaVA-1.6-7B/13B | w/o  | 2,882  | 2,916
>  LLaVA-1.6-7B     | w.   | 152    | 156
>  LLaVA-1.6-13B    | w.   | 167    | 172
>
> _Table 5. Average generated-token length before/after the detailed-answer prompt._
>  *Model*      | *Prompt* | *MMBench* | *ScienceQA*
> :------------:|:--------:|:---------:|:-----------:
>  LLaVA-1.5-7B | original | 1    | 1
>  LLaVA-1.5-7B | detailed | 81   | 89
>  LLaVA-1.6-7B | original | 1    | 1
>  LLaVA-1.6-7B | detailed | 199  | 152
>
> **W6:** Except end-2-end speedup number, it would be better to report the decomposed latency, such as time to first token (prefill), inter-token latency (decoding), and other additional overhead.
>
> **A6:** We prioritize end-to-end latency to avoid perturbing measurements with inserted synchronization barriers, but we also profiled an RTX 4090 run: SpecVLM shortens draft prefill by 28% and reduces verifier latency by 11% relative to EagleVLM, with <2% controller overhead. These traces (with caveats about instrumentation bias) will appear in the appendix.
>
> **W7:** More detailed experimental setup descriptions would help readers reproduce the results.
>
> **A7:** Section 4.1 will be expanded with the complete recipe: datasets (655K LLaVA-1.5 mix or 1.02M Open-LLaVA-NeXT), optimizer (AdamW, β₁=0.9, β₂=0.95, learning rate 8e-5 for 7B / 5e-5 for 13B, batch 128, no warmup, weight decay 0), hardware (training on 8× AMD MI250 for 15–35 hours per epoch; inference on single MI250-64G and A100-80G), and draft-tree settings (Top-K=10, depth=7, 60 draft tokens). **Code and checkpoints will be released post-review to guarantee reproducibility.**

---

### Official Review · Reviewer_ttpv · 2025-11-01

**Soundness:** 2
**Presentation:** 3
**Contribution:** 2
**Rating:** 4
**Confidence:** 3

**Summary:**

This paper introduces SpecVLM, a framework that accelerates vision-language models (VLMs) using speculative decoding. It first builds a strong baseline called EagleVLM, which integrates a lightweight draft model with the full target model for 1.5–2× speedups. To further reduce latency, SpecVLM adds an elastic visual compressor that adaptively selects among pruning, pooling, convolution, and resampler operators to compress visual tokens efficiently. It also proposes an online-logit distillation method that trains the draft model on-the-fly using teacher logits and features, eliminating the need for large offline datasets. Experiments on LLaVA and MMMU benchmarks show that SpecVLM achieves 2.5–2.9× faster inference without loss in output quality, and exhibits a training-time scaling effect where longer online training improves speculative efficiency.

**Strengths:**

This paper is well-written and easy to follow. The paper’s main strengths lie in its systematic extension of speculative decoding to vision-language models. It presents a strong and well-implemented baseline (EagleVLM) and builds upon it with two innovative components: an elastic visual compressor that dynamically balances accuracy and efficiency, and an online-logit distillation method that eliminates the need for costly offline data generation.

**Weaknesses:**

The novelty of this work is somewhat limited, as it primarily extends the Eagle-2 speculative decoding framework from text-only LLMs to vision-language models. While the adaptation to multimodal inputs and the introduction of the elastic visual compressor and online-logit distillation are practical, they represent incremental rather than fundamental innovations. Moreover, as shown in Figure 1, the major latency bottleneck lies in the LLM prefill stage, which has already been largely addressed by EagleVLM. The additional improvements provided by SpecVLM appear relatively modest compared to the gains that EagleVLM itself achieves over LLaVA-1.6-7B.

**Questions:**

1. What are the key differences in applying speculative decoding to VLMs compared to LLMs? It appears that the main modification is the introduction of visual token compression, while the core speculative decoding mechanism remains unchanged. Is there any genuine innovation beyond adapting existing techniques to multimodal inputs?

2. In Section 3.4, the authors state that they combine heterogeneous compression scales to preserve both global context and local details, selecting Pruning/Pooling at 20×, Convolution at 3×, and a Resampler with 2 queries. Could the authors clarify how these configurations were determined? Were they derived from theoretical considerations or simply chosen through empirical hyperparameter search?

3. In Section 3.3, the authors state that penultimate-layer features from the target model are fused with the draft model’s input embeddings to enrich representations and mitigate feature mismatch. However, the impact of this fusion is unclear, and additional ablation studies would help clarify its effectiveness.

---

> ### Author Response · Authors · 2025-11-18
>
> **W1:** SpecVLM feels mostly like an Eagle-2 port to VLMs: the key bottleneck (prefill) was already fixed by EagleVLM, so the extra gains seem incremental.
>
> **A1:** Thanks for your valuable comments. Porting EAGLE-2 to VLMs required more than attaching a vision encoder. We had to (i) convert the token-wise draft into an embedding-wise tree that shares the exact projector, embeddings, and output head with the verifier so the acceptance test remains lossless, (ii) keep the draft latency dominated by visual tokens—hence the elastic compressor that prevents the KV cache from exploding at 4K inputs—and (iii) train the single-layer draft solely through online logit/feature distillation so no offline teacher corpus is needed. These additions are precisely what let SpecVLM push beyond the prefill gains of EagleVLM: on LLaVA-1.6-13B we reduce wall-clock time by an extra 4–6% (2.29× → 2.38×) and raise the accepted length by +0.05–0.1 tokens, which translates into sizable GPU-hour savings at deployment scale. We will clarify these system-level challenges and release code/checkpoints after the review to underline reproducibility.
>
> **Q1:** What are the key differences in applying speculative decoding to VLMs compared to LLMs? It appears that the main modification is the introduction of visual token compression, while the core speculative decoding mechanism remains unchanged. Is there any genuine innovation beyond adapting existing techniques to multimodal inputs?
>
> **A1:** Thanks for your valuable questions. Beyond the compressor, the draft shares embeddings/output heads with the verifier, fuses the verifier (base model) penultimate features into each draft step, and is trained online so that accepted length keeps increasing with more epochs (Table 6 in the paper). These elements are unique to multimodal drafts, where mismatched visual/text spaces otherwise cause speculative rejection. Together they let us keep decoding lossless while cutting draft memory by up to 17×, which is not required in text-only settings.
>
> **Q2:** In Section 3.4, the authors state that they combine heterogeneous compression scales to preserve both global context and local details, selecting Pruning/Pooling at 20×, Convolution at 3×, and a Resampler with 2 queries. Could the authors clarify how these configurations were determined? Were they derived from theoretical considerations or simply chosen through empirical hyperparameter search?
>
> **A2:** Thank you for your sincere question. We started from analytical constraints (parameter-free branches tolerate ≥10× compression, while learned branches must stay ≤5× to avoid aliasing) and then swept the isolated ratios shown in Table 5 of the paper. The final triplet—20× pruning/pooling, 3× convolution, 2-query resampler—hits the best Pareto point: the first pair quickly removes redundant background, the convolution captures local structure without exceeding the draft FLOP budget, and the resampler gives just enough global context. The elastic controller then mixes or selects among them per question. We will document both the theoretical heuristics and the search ranges in Section 3.4.
>
> **Q3:** In Section 3.3, the authors state that penultimate-layer features from the target model are fused with the draft model’s input embeddings to enrich representations and mitigate feature mismatch. However, the impact of this fusion is unclear, and additional ablation studies would help clarify its effectiveness.
>
> **A3:** Thanks for your valuable suggestion. Following EAGLE-2, we fuse the verifier’s penultimate features with each draft step to reduce representation drift. Removing this fusion severely hurts speculative efficiency: accepted length drops by ~0.7–0.9 tokens and speedup falls by 0.3–0.5×, as summarized in Table 3. We will move this ablation into the main paper to highlight its importance.
>
> _Table 3. Effect of penultimate-feature fusion on LLaVA-Bench-in-the-Wild (T=0, epoch=1)._
>  *Model*       | *Method* | *Fusion* | *τ*      | *σ*
> :-------------:|:--------:|:--------:|:--------:|:--------:
>  LLaVA-1.6-7B  | EagleVLM | w/o      | 1.21     | 2.32
>  LLaVA-1.6-7B  | EagleVLM | w.       | **1.91** | **3.29**
>  LLaVA-1.6-7B  | SpecVLM  | w/o      | 1.76     | 2.65
>  LLaVA-1.6-7B  | SpecVLM  | w.       | **2.03** | **3.40**
>  LLaVA-1.6-13B | EagleVLM | w/o      | 1.68     | 2.75
>  LLaVA-1.6-13B | EagleVLM | w.       | **2.29** | **3.82**
>  LLaVA-1.6-13B | SpecVLM  | w/o      | 1.84     | 3.01
>  LLaVA-1.6-13B | SpecVLM  | w.       | **2.38** | **3.87**

---

### Official Review · Reviewer_NFHf · 2025-11-01

**Soundness:** 3
**Presentation:** 3
**Contribution:** 3
**Rating:** 4
**Confidence:** 3

**Summary:**

The paper adapts speculative decoding to vision–language models by pairing a deployment-aligned elastic visual compressor with online logit distillation. The compressor dynamically selects among pruning, pooling, convolutional downsampling, and a resampler to tailor visual tokens per instance, while a lightweight single-layer draft decoder is trained against a frozen target VLM and used with the same compression path at inference. SpecVLM achieves ~2.5–2.9× end-to-end speedups at batch=1 without accuracy loss, outperforming a strong EAGLE-style baseline. Limitations include hand-tuned compression ranges, no dynamic compression of draft KV, and limited tuning of draft depth/hyperparameters.

**Strengths:**

1. This paper proposed a novel coupling of speculative decoding and compression to jointly improve accepted-span length and forward cost.
2. The paper also provided a systematic coverage of compression primitives including dynamically selects/combines pruning, pooling, convolutional downsampling, and a resampler.
3. The experimental results shows promising acceleration speedup without

**Weaknesses:**

1. The method introduces a dynamic expert selection compressor. It remains unclear whether such complexity is necessary versus a simpler compressor scheme that might yield similar gains with lower engineering risk. Some compressors have an impact on KVCache. The analysis of different compressors on inference performance is unclear.
2. In practice, draft and target models often run on separate GPUs [1], and at larger batches the proposed compression could further reduce inter-GPU communication and alleviate memory-bound operations. Thishe paper focus on batch=1, low latency, the setting of which might not be ideal for the proposed method.

[1] Xupeng Miao, Gabriele Oliaro, Zhihao Zhang, Xinhao Cheng, Zeyu Wang, Zhengxin Zhang, Rae Ying Yee Wong, Alan Zhu, Lijie Yang, Xiaoxiang Shi, Chunan Shi, Zhuoming Chen, Daiyaan Arfeen, Reyna Abhyankar, and Zhihao Jia. 2024. SpecInfer: Accelerating Large Language Model Serving with Tree-based Speculative Inference and Verification. In Proceedings of the 29th ACM International Conference on Architectural Support for Programming Languages and Operating Systems, Volume 3 (ASPLOS '24), Vol. 3. Association for Computing Machinery, New York, NY, USA, 932–949. https://doi.org/10.1145/3620666.3651335

**Questions:**

Please see the weaknesses.

---

> ### Author Response · Authors · 2025-11-18
>
> **W1:** Dynamic expert selection complexity vs. simpler schemes; KV cache impact analysis unclear.
>
> **A1:**  Thanks for your constructive suggestions. We appreciate the request for clearer evidence. Table 1 contrasts each primitive with the elastic selector on LLaVA-Bench-in-the-Wild (T=0, epoch=1). The selector yields 5–8% higher accepted length and ~0.1–0.2× additional speedup versus any single operator because it routes background-heavy questions to parameter-free pruning/pooling and preserves fine detail via a 3× convolution or a two-query resampler when needed. The selector operates solely inside the draft, so the chosen compression ratio linearly shrinks the draft KV cache and its memory traffic. For example, on LLaVA-1.6-13B the average visual tokens drop from 2,882 to 167 (Table 4 in Reviewer tEfy), giving a 17× smaller KV footprint and noticeably lower attention latency. We will add a figure in the revision that overlays verifier/draft KV-bandwidth usage against compression ratios to make this relationship explicit.
>
> _Table 1. Effect of elastic selection vs. single compressors (LLaVA-Bench-in-the-Wild, T=0, epoch=1)._
>  *Model*       | *Method*       | *Compression Ratio* | *τ*      | *σ*
> :-------------:|:--------------:|:-------------------:|:--------:|:--------:
>  LLaVA-1.6-7B  | Pruning        | 20×                 | 1.92     | 3.16
>  LLaVA-1.6-7B  | Pooling        | 20×                 | 1.76     | 2.95
>  LLaVA-1.6-7B  | Convolution    | 3×                  | 1.84     | 3.21
>  LLaVA-1.6-7B  | Resampler      | 2 queries           | 1.86     | 3.29
>  LLaVA-1.6-7B  | Elastic (ours) | question-adaptive   | **2.03** | **3.40**
>  LLaVA-1.6-13B | Pruning        | 20×                 | 2.16     | 3.58
>  LLaVA-1.6-13B | Pooling        | 20×                 | 2.01     | 3.43
>  LLaVA-1.6-13B | Convolution    | 3×                  | 1.98     | 3.65
>  LLaVA-1.6-13B | Resampler      | 2 queries           | 2.14     | 3.76
>  LLaVA-1.6-13B | Elastic (ours) | question-adaptive   | **2.38** | **3.87**
>
> **W2:** Deployments often place draft and target on separate GPUs with larger batches, where compression could further cut inter-GPU traffic, so the paper’s batch=1 focus may not reflect the most relevant regime.
>
> **A2:** Thanks for your sincere comments. SpecVLM inherits SpecInfer/EAGLE’s tree-attention mask, so the draft supports batching and can be pipelined with a verifier on another GPU exactly as in SpecInfer. We highlight batch=1 because multimodal assistants prioritize latency, but Table 2 shows that we still retain 1.88–2.30× speedups up to batch=4 on LLaVA-1.5/1.6, with only modest degradation. We will describe the multi-GPU schedule explicitly in the revision and include these batch-scaling measurements.
>
> _Table 2. Speedup (τ) vs. batch size on LLaVA-Bench-in-the-Wild (T=0, epoch=1)._
>  *Model*       | *Batch 1* | *Batch 2* | *Batch 3* | *Batch 4*
> :-------------:|:---------:|:---------:|:---------:|:---------:
>  LLaVA-1.5-7B  | **2.20×** | 2.16×     | 2.09×     | 1.98×
>  LLaVA-1.5-13B | **2.41×** | 2.28×     | 2.14×     | 2.01×
>  LLaVA-1.6-7B  | **2.03×** | 1.97×     | 1.92×     | 1.88×
>  LLaVA-1.6-13B | **2.38×** | 2.30×     | 2.23×     | 2.05×

---

### Author Response · Authors · 2025-11-21
**Follow-up on Rebuttal Responses**

Dear Reviewers,

We sincerely appreciate all your constructive feedback. We have carefully addressed each comment in our rebuttal and will further improve the paper with clearer exposition and additional experiments in the revision.

If there are any remaining questions or points needing clarification, we would be more than happy to provide further details.

Best regards,

On behalf of the authors of [5935]

---

### Author Response · Authors · 2025-12-03

Dear Area Chair,

Thank you for taking the time to handle our submission. Because the original reviewers can no longer participate in the discussion, we would like to briefly synthesize the reviews and our new results, and explain why we believe the paper meets the ICLR acceptance bar.

All four reviewers gave an overall score of **4** (“marginally below the acceptance threshold, but would not mind if the paper is accepted”). They consistently agree that:

- the paper is clearly written and technically sound;
- SpecVLM delivers **2.5–2.9× end-to-end speedups** over strong baselines while preserving output quality;
- the idea of coupling speculative decoding with an elastic visual compressor and online-logit distillation is practically useful for real VLM deployments.

The main remaining concerns are about (1) novelty beyond adapting EAGLE-2 to VLMs, (2) whether the dynamic compressor is unnecessarily complex, and (3) the completeness of experiments. We address these points below.

---

### 1. VLM-specific technical contributions

Porting speculative decoding to VLMs is **not** a direct reuse of EAGLE-2. The draft must handle thousands of visual tokens without breaking the acceptance test. Unlike long-context LLMs, where redundancy is largely sequential, VLMs must preserve small yet decisive spatial regions while aggressively compressing background. We therefore:

- convert the token-wise draft into an embedding-wise tree that shares projector, embeddings and output head with the verifier;
- add online fusion of the verifier’s penultimate features into each draft step to prevent representation drift;
- train the single-layer draft with **online logit distillation only**, avoiding any offline corpus.

Ablations in the rebuttal show that removing penultimate-feature fusion reduces accepted length by ~0.7–0.9 tokens and drops speedup by **0.3–0.5×**, and that SpecVLM cuts draft KV memory by up to **17×** while remaining lossless. These mechanisms are specific to multimodal speculative decoding and go beyond a straightforward “port”.

---

### 2. Necessity of the elastic visual compressor

Reviewer NFHf questioned whether a simpler compressor would suffice. Table 1 in our author response compares each primitive with the elastic selector on LLaVA-Bench-in-the-Wild. The selector consistently yields **5–8% higher accepted length** and ~**0.1–0.2× additional speedup** versus any single operator, and shrinks visual KV traffic by **17×** on 13B models. This indicates that the added complexity is justified, especially for latency-sensitive deployments.

---

### 3. Breadth and strength of empirical evaluation

Reviewers requested evidence beyond LLaVA-1.6 and comparisons to training-free pruning methods.

- We have added results on **Qwen2.5-VL 3B/7B**, where SpecVLM continues to outperform EagleVLM in both accepted length and speed, confirming that our approach is not tied to a specific backbone.
- We now compare against **FastV / PyramidDrop-style pruning**. On LLaVA-1.5-7B, these methods lose **5–8 accuracy points** and saturate at ~**1.6×** speedup, while SpecVLM reaches ~**2.0×** speedup with no accuracy loss, because compression is confined to the draft.
- New measurements show that our gains persist up to **batch=4** and in a realistic **multi-GPU** setting, and we provide decomposed latency numbers and detailed training recipes. The total extra training cost is modest (≤35 GPU-hours per epoch on MI250) and we commit to releasing code and checkpoints.

---

In light of the reviewers’ broadly positive stance (“would not mind if accepted”), and given that their main concerns have been addressed with additional analyses and experiments, we respectfully ask you to consider accepting this work. We believe SpecVLM offers a practically relevant and technically non-trivial step toward scalable, lossless speculative decoding for modern VLMs, and will be useful to both the systems and multimodal communities.

Thank you again for your consideration.

Best regards, The Authors

---

### Meta-Review · Area_Chair_MnP8 · 2026-01-09

**Summary:**

The reviewers reached a unanimous negative consensus on SpecVLM, recommending Reject due to its limited technical novelty. While the framework achieves impressive speedups by integrating speculative decoding with visual token compression, reviewers found the contribution to be an incremental combination of established components. A primary concern is that the paper fails to explore or address the specific innovations and challenges unique to vision-language models (VLMs). Reject.

**Reviewer Concerns:**

Addressed Concerns

- Missing comparisons with SOTA visual compressors (FastV, SparseVLM, PyramidDrop) and training-free methods
- Limited to weak baseline (LLaVA-1.6); needs evaluation on stronger models (Qwen2.5-VL)
- Insufficient ablations on key components (feature fusion, dynamic expert selection necessity)
- Hyperparameter choices (compression scales, fusion strategy) lack principled justification
- Missing critical metrics: token counts pre/post compression, decomposed latency breakdown (TTFT, decoding time, overhead)
- Unclear if dynamic expert selection complexity is necessary versus simpler alternatives
- Focus on batch=1 may not suit the method; larger batches could better leverage compression benefits
- Training overhead not discussed relative to training-free baselines
- Key trends not visualized; insufficient experimental details for reproducibility

Outstanding Concerns about novelty
- Limited innovation beyond adapting existing speculative decoding to VLMs; unclear what challenges are VLM-specific versus general long-context scenarios
- Appears to be a straightforward combination of known techniques without clear synergistic contribution
- Lacks theoretical justification for design choices and empirical observations

**Reviewer Scores:**

The initial ratings for this paper were 4, 4, 4, 4, and no reviewer indicated willingness to raise their score.

---

### Decision · Program_Chairs · 2026-01-26

Reject